# Pneumococcal nasopharyngeal carriage in Indonesia infants and toddlers post-PCV13 vaccination in a 2+1 schedule: A prospective cohort study

Ari Prayitno[1]*, Bambang Supriyatno[1], Zakiudin Munasir[1], Anis Karuniawati[2], Sri Rezeki S. Hadinegoro[1], Joedo Prihartono[3], Dodi Safari[4], Julitasari Sundoro[5], Miftahuddin Majid Khoeri[4]

1 Department of Pediatrics, Faculty of Medicine Universitas Indonesia – Cipto Mangunkusumo Hospital, Jakarta, Indonesia, 2 Department of Microbiology, Faculty of Medicine Universitas Indonesia, Jakarta, Indonesia, 3 Department of Community Medicine, Faculty of Medicine Universitas Indonesia, Jakarta, Indonesia, 4 Molecular Bacteriology, Eijkman Institute for Molecular Biology, Jakarta, Indonesia, 5 Indonesian Technical Advisory Group on Immunization, Jakarta, Indonesia

* ariprayitno@yahoo.com

**Data Availability Statement:** All relevant data are within the manuscript and its Supporting information files.

## Abstract

### Background

The PCV13 immunization demonstration program began in October 2017 in Indonesia. The aim of this study is to assess the dynamic changes of pneumococcal serotype before and after PCV13 administration, with two primary and one booster doses.

### Methods

The prospective cohort study was conducted as a follow up study measuring the impact of PCV13 demonstration program by the Indonesian Ministry of Health in Lombok Island, West Nusa Tenggara, Indonesia, from March 2018 to June 2019. The subjects were two-month-old healthy infants who were brought to the primary care facility for routine vaccination and followed until 18 months of age. We use convenience sampling method. There were 115 infants in the control group and 118 infants in the vaccine group, and the PCV immunization was given on a 2+1 schedule. Nasopharyngeal (NP) swabs were collected four times during the vaccination periods by trained medical staff. Specimens were analyzed by culture methods to detect *S. pneumonia* colonization and multiplex polymerase chain reaction (mPCR) to determine serotype. The most frequently detected serotypes will be named as dominant serotypes. Descriptive analysis of demographic characteristics, the prevalence of overall and serotype colonization, and the distribution of serotypes were performed. The prevalence of both cohort groups were compared using chi-square test. Statistical significance was set at p < 0.05.

### Results

Two hundred and thirty three infants age two months old were recruited, with 48.9% of the subjects were male and 51.1% of the subjects were female. Sociodemographic data in both

**Funding:** The authors received no specific funding for this work.

**Competing interests:** The authors have declared that no competing interests exist.

cohort groups were relatively equal. Nasopharyngeal pneumococcal colonization before PCV13 administration occurred in 19.1% of the control and 22.9% of the vaccine group. The prevalence increased with increasing age in both groups. The prevalence of VT serotypes in control groups aged 2 months, 4 months, 12 months, and 18 months was 40.9%, 44.2%, 53.8%, and 54.3%, respectively, and in the vaccine group, 25.9%, 40.4%, 38.0%, and 22.6%, respectively. The most common VT serotypes in both groups were 6A/6B, 19F, 23F, and 14. The prevalence of VT serotypes decreased significantly compared to non-vaccine type serotypes after three doses of the PCV13 vaccine ($p < 0.001$). Another notable change was the decline in prevalence of serotype 6A/6B after PCV13 administration using the 2+1 schedule.

## Conclusions

This study shows lower prevalence of VT and 6A/6B serotypes in the nasopharynx among children who were PCV13 vaccinated compared with those who were unvaccinated. The result from this study will be the beginning of future vaccine evaluation in larger population and longer period of study.

## Introduction

Pneumonia is the leading cause of death in children under five years old worldwide [1]. More than two million children die from pneumonia every year, which exceeds the total number of deaths caused by AIDS, malaria, and measles combined. Worldwide, one out of five deaths of children under five years old is caused by pneumonia. The Indonesian Health Profile reported that 922,000 Indonesian children under five years of age died from pneumonia in 2015 and West Nusa Tenggara is in first place with total cases in children under five was 33,291 cases [2, 3].

The most common cause of bacterial pneumonia is *Streptococcus pneumoniae*, which lives in colonies in the human nasopharynx. The transmission of various pneumococcal infections can be prevented by the administration of the pneumococcal conjugate vaccine in infants under one year of age. PCV10 dan PCV13 are licensed to reduce invasive pneumococcal disease (IPD). Two doses of PCV10 or PCV13 given before the age of 12 months at an interval of 4–8 weeks or more and one booster doses given between 9–15 months of age [4]. The effectiveness of PCV is important because it correlates with a reduced disease burden of IPD causes by vaccine-type (VT) serotype. Harboe et al. [5] showed that IPD incidence was reduced 21% in the total population and 71% in children under five years old after PCV13 administration in the Netherlands. PCV reduces carriage, and thereby transmission of VT pneumococci, providing both direct and indirect protection from common disease-causing serotypes. As a PCV program matures, VT serotype may reduce and replacement may occur by non-vaccine type serotype (NVT) [6, 7].

In Indonesia, research on the nasopharyngeal colonization in children was limited. The earliest pneumococcal study conducted on Lombok Island in 1997 and followed up in 2012 with PCV13 administration. The results of both studies preceding PCV and after use of PCV13 showed no difference in the prevalence of nasopharyngeal colonization of *S. pneumoniae* (48% in 1997 and 46% in 2012) [8, 9]. Other study conducted in Bandung City and semi-rural area in Padalarang district (~25km from Bandung City), regarding pneumococcal carriage

prevalence, mentioned the prevalence was 22.0% at the beginning of the study and increased to 68.4% after 10 months from study [10].

In 2012, the World Health Organization Strategy Advisory Group of Experts (WHO SAGE) recommended the administration of PCV for national immunization programs world-wide. Specifically, the WHO recommended three doses of PCV, either three primary doses (3 +0) or two primary doses plus booster (2+1). There are multiple factors to consider in implementing a national immunization program, such as the interval of primary doses, the use of booster, and cost-effectiveness. There is no single schedule that is optimal for every setting. According to WHO, the 3+0 vaccination schedule of PCV may be preferred in countries in which disease rates peak well before the end of the first year of life or when the immunization coverage is low if given after the first year, whereas the 2+1 schedule is given when the duration of protection becomes a concern or for specific protection against serotypes 1, 3, or 5. Under the observation of an invasive pneumococcal disease, the 2+1, 3+0, and 3+1 schedules are equally effective. In addition, PCV immunization will help to increase immunization coverage if the schedule coincides with the established national immunization schedule, as it will be more convenient both for the parents and health workers [11].

Indonesia requires an optimal schedule to gain maximum protection for vulnerable age groups. Since 2017, the national PCV immunization program conducted in Indonesia has been via the demonstration program, in which PCV immunization begins in one small area then gradually expands to the entire country. The immunization schedule used is 2+1, adjusted to the national immunization schedule. The first dose of PCV13 is administered to two-month-old children simultaneously with DPT/HepB/Hib1 and OPV2, and the second dose of PCV13 is administered to children three months old along with DPT/HepB/Hib1 and OPV3. The third PCV13 injection (booster) is given to children 12 months old. Evaluation of the program effectiveness is necessary as the basis for continuing the PCV immunization program to other provinces. Pneumococcal vaccine effectiveness can be assessed by measuring changes in *S. pneumoniae* serotype colonization in the nasopharynx, which compares the prevalence of VT serotypes to NVT serotypes.

The goal of this research is to assess the dynamic changes of *S. pneumoniae* colonization before and after PCV13 administration using two primary doses (in a one-month interval) and one booster dose in healthy infants and toddlers aged 2–18 months in Indonesia. After vaccination with PCV13, we hypothesized that VT serotypes would be replaced by NVT serotypes or become no colonization.

## Methods

### Study design and participants

A prospective cohort study was conducted as a follow up study measuring the impact of PCV13 demonstration program in Lombok Island, West Nusa Tenggara, Indonesia, from March 2018 until June 2019. In 2017, total population of under five children in Lombok Island was 290,866 with estimated 18,557 children with pneumonia (6.38%). The administration of PCV13 on a 2+1 schedule was carried out by the Indonesian Ministry of Health as demonstration program in Lombok (HK.01.07 / MENKES / 199 / 2017). The Ministry of Health administered PCV13 when subjects 2, 3, and 12 months old, whereas this study obtained nasopharyngeal swab before the first dose of PCV13, at the age of 4 months old, before booster dose administration at 12 months old, and 6 months after the last dose. The demonstration program started in 2017 from 2 districts in West Nusa Tenggara Province (West Lombok and East Lombok as the vaccine group) to 251 infants, where the most

cases of pneumonia were found, and expanded in 2018 to the entire Lombok and Bangka district.

The study has been reviewed and approved by ethical committee of Faculty of Medicine, Universitas Indonesia, Jakarta, Indonesia (1053/UN2.F1/ETIK/2017). Written informed consent was obtained from parents/guardians of the infants before the study. In addition, parents/guardians were also asked for the full information of address and phone number of parents or guardian as follow-up methods to encourage them to come when the infants were 4, 12, and 18 months old.

The eligibility criteria for subjects of this study were two-month-old healthy infants who were brought to the primary care facility for routine vaccination in Lombok Island. Medical history, demographics data, living conditions, vital signs assessment, and anthropometric measurement were recorded on a case report form before the vaccination given. We use convenience sampling method with at least 90 subjects per group based on sample size ($n$) formula for cohort studies $n = \frac{(Z\alpha\sqrt{(1+1/m)p(1-p)}+Z\beta\sqrt{\wp0(1-\wp0)/m+\wp1(1-\wp1)})^2}{(\wp0-\wp1)^2}$, with $p = \frac{\wp1+m\wp0}{m+1}$. Here, $Z\alpha$ is standard normal variate for level of significance ($\alpha$ = 5%; $Z\alpha$ = 1.96), m = number of control subject per vaccinated subject (m = 1), $Z\beta$ is standard normal variate for power or type 2 error ($\beta$ = 80%; $Z\beta$ = 0.84), p is the average proportion exposed, $\wp0$ is the probability of events in control group based on previous research prevalence (26%) [9], and $\wp1$ is the probability of events in vaccinated group based on previous research (10%) [12, 13]. Ten percent of drop out estimation ($f$) was calculated using formula $n' = \frac{n}{(1-f)}$, then the minimum subjects for each group are 110 subjects. Subjects lived in Central Lombok have not immunized for PCV at the mean time and were enrolled to control group (n = 115), whereas subjects lived in West or East Lombok received PCV13 immunization (as the Indonesian Ministry of Health program) and were enrolled to vaccine groups (n = 118).

The exclusion criteria were infants with acute infection (such as fever, acute otitis media, respiratory tract illness) within 7 days of recruitment, chronic illness, immunocompromise, or received antibiotic treatment during the 3 days before recruitment. Control and vaccine groups were age and location matched. If a recruited child became unwell over the period they aged from 2 months to 18 months, the nasopharyngeal swab would be performed after the catch-up immunization.

## Sample collection

Nasopharyngeal (NP) swab specimens were collected from children by trained medical staff using a flexible nasopharyngeal flocked nylon swab (Copan, Italy; #503SC01), as recommended by WHO [14]. We collected four swabs during the vaccination periods: before vaccination (two months of age; the first dose), one month after the second dose (four months of age), before the third dose (12 months of age), and six months after the booster (18 months of age). The NP swabs were placed in cryotubes containing 1 mL skimmed milk, tryptone, glucose, and glycerol (STGG) as a transport medium and placed in a cooler with icepacks. Within 4 hours of collection, the inoculated STGG samples were vortexed for 10–20 sec prior to storage in a −80˚C freezer at the Biomedical Laboratory General Hospital of West Nusa Tenggara Province [15, 16]. All NP-STGG specimens were transported in dry ice to the Eijkman Institute for Molecular Biology, Jakarta, Indonesia, for further analysis.

## Laboratory procedures

Isolation and identification of *S. pneumoniae* were performed using methods previously described [17]. Briefly, 100 μL aliquots of each NP-STGG sample were added to 5 mL

Todd-Hewitt broth supplemented with 0.5% yeast extract and 1 mL of rabbit serum, and the mixture was incubated for 5 hours at 37˚C in a $CO_2$ incubator. We cultured 10 μL of enriched NP swab on a sheep blood agar plate (BAP), which was incubated at 37˚C with 5% $CO_2$ for 18–20 hours. The colonies with pneumococcal characteristics (watery, alpha-hemolytic, flat-depressed center) were sub-cultured on BAP. An optochin disk (30 μg) was then placed on the surface of the inoculated blood agar plate for optochin susceptibility testing. The colonies susceptible to optochin (diameter > 14 mm) were defined as *S. pneumoniae* [15]. We performed bile solubility testing to confirm colonies with pneumococcal characteristics but with optochin resistance. Pneumococcal isolates were serotyped by conventional multiplex PCR. The microbiologists undertaking laboratory procedures were blinded to participant cohort and demographics [18].

## Data analysis

Data from case report forms and laboratory was entered to Microsoft Excel 365. Datasets were imported and statistical analyzed using Graphpad Prism 8. Descriptive analysis of demographic characteristics, the prevalence of overall and serotype colonization, and the distribution of serotypes was performed in the beginning of the study to determine if control and vaccine groups were different at baseline. The differentiation at baseline may leads to different outcomes. Categorical variables were summarized by counts and percent. Inferential analyses using the chi-square test and determining the odds ratio were performed to compare data between control and vaccine group. Statistical significance was assigned at $p < 0.05$. However, this study will not be assessing vaccine effectiveness due to small sample size and limited time.

PCV13 serotypes 1, 3, 4, 5, 6A, 6B, 7F, 9V, 14, 18C, 19A, 19F, and 23F are considered vaccine-type (VT) serotypes, while others are non-vaccine type (NVT) serotype. If an isolate was detected, but typing was not possible due to biological reasons or technical issues, isolate will be assigned as untypeable serotype which included in NVT serotype. If multiple pneumococcal serotypes are identified, the infant will be recorded and observed based on the number of serotypes. Therefore, there will be higher number of serotypes than the number of infants. However, the data of each infants on carrying how many serotypes and the changes will not be reported in this study due to some considerations. The most frequently detected serotypes will be named as dominant serotypes. The following variables were assessed: gender, nutritional status, primary health care area, number in household, and cigarette smoke exposure as confounding variables. Nutritional status was assessed according to WHO guidelines (weight-for-age Z-score).

## Results

### Demographic characteristic

The first nasopharyngeal swab was collected from 233 two-month-old infants (115 subjects in control groups and 118 subjects in vaccine groups). The second and third nasopharyngeal swabs were obtained from 230 four-month-old infants and 223 12-month-old infants, respectively. The last nasopharyngeal swab was collected from 233 18-month-old children. Ten infants dropped out of the study, six infants moved, three infants refused to continue the study, and one died of febrile convulsions (Fig 1). All infants were included in the analysis as originally allocated to control and vaccine groups.

Gender prevalence in both groups were relatively equal: in the control group, 46.1% were male, and 53.9% were female, and in the vaccine group, 51.7% were male, and 48.3% were female. Most commonly, subjects lived in a household of 4–6 people (63.5% in the control group and 68.6% in the vaccine group). Most 2-month-old infant subjects were exposed to

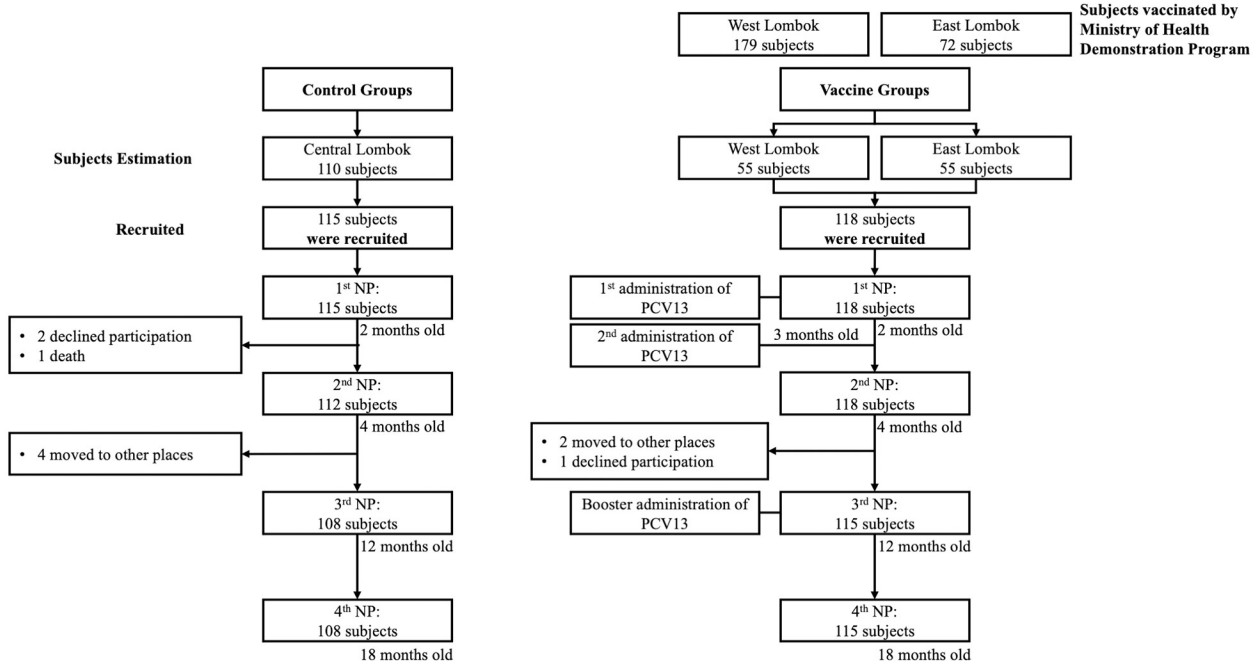

**Fig 1. Flow chart of participants in the study.**

cigarette smoke (99 of 115 [86.1%] in the control group and 92 of 118 [78%] in the vaccine group). However, there was a significant difference of nutritional status between cohort groups (p < 0.001). Most subjects in the control group (106 of 115 subjects, 92.2%) and the vaccine group (86 of 122 subjects, 72.9%) had good nutritional status (Table 1).

Table 2 shows the demographic characteristics in *S. pneumoniae* positive and negative groups. There were significant differences in gender between the positive and no colonization groups, yet other demographic characteristics were not significantly different.

**Table 1. Demographic characteristics by cohort groups.**

| Characteristics | Control | | | Vaccine | | | Overall *p*-value |
|---|---|---|---|---|---|---|---|
| | n (%) | Odds Ratio (95% CI) | *p*-value | n (%) | Odds Ratio (95% CI) | *p*-value | |
| **Gender** | | | | | | | 0.392 |
| Male | 53 (46.1) | 2.42 (0.93–6.34) | 0.066 | 61 (51.7) | 2.77 (1.10–6.98) | 0.027 | |
| Female | 62 (53.9) | Reference | | 57 (48.3) | Reference | | |
| **Nutritional Status** | | | | | | | < **0.001** |
| Overweight + Obese | 9 (7.8) | N/A | 0.129 | 21 (17.8) | N/A | 0.003 | |
| Normal weight | 106 (92.2) | Reference | | 86 (72.9) | Reference | | |
| Underweight | 0 (0) | N/A | N/A | 11 (9.3) | N/A | 0.029 | |
| **Number in household** | | | | | | | 0.123 |
| 1–3 | 27 (23.5) | 0.34 (0.07–1.81) | 0.195 | 16 (13.6) | 1.07 (0.24–4.84) | 0.933 | |
| 4–6 | 73 (63.5) | 0.71 (0.19–2.55) | 0.600 | 81 (68.6) | 0.91 (0.29–2.84) | 0.880 | |
| ≥ 7 | 15 (13) | Reference | | 21 (17.8) | Reference | | |
| **Cigarette smoke exposure** | | | | | | | 0.107 |
| Yes | 99 (86.1) | 4.08 (0.50–32.35) | 0.158 | 92 (78.0) | 0.46 (0.18–1.19) | 0.107 | |
| No | 16 (13.9) | Reference | | 26 (22.0) | Reference | | |

**Table 2. Demographic characteristics in *S. pneumoniae* positive and negative groups.**

| Characteristics | *S. pneumoniae* | | Odds Ratio *(95% CI)* | *p*-value[a] |
|---|---|---|---|---|
| | Positive (n = 49) | No Colonization (n = 184) | | |
| Gender | | | | **0.004** |
| Male | 33 (28.9%) | 81 (71.1%) | 2.62 (1.35–5.09) | |
| Female | 16 (13.4%) | 103 (86.6%) | Reference | |
| Nutritional Status | | | | 0.255 |
| Overweight+Obese | 0 | 30 (100%) | N/A | |
| Normal weight | 49 (25.5%) | 143 (74.5%) | N/A | |
| Underweight | 0 | 11 (100%) | N/A | |
| Primary Health Care Area | | | | 0.261 |
| Praya | 22 (19.1%) | 93 (80.9%) | 0.99 (0.36–2.69) | |
| Gerung | 10 (18.2%) | 45 (81.8%) | 0.93 (0.30–2.85) | |
| Lenek | 11 (34.4%) | 21 (65.6%) | 2.18 (0.69–6.90) | |
| Selong | 6 (19.4%) | 25 (80.6%) | Reference | |
| Number in household | | | | 0.625 |
| 1–3 | 7 (16.3%) | 36 (83.7%) | 0.58 (0.19–1.76) | |
| 4–6 | 33 (21.4%) | 121 (78.6%) | 0.82 (0.35–1.91) | |
| $\geq 7$ | 9 (25%) | 27 (75%) | Reference | |
| Cigarette smoke exposure | | | | 0.625 |
| Yes | 39 (20.4%) | 152 (79.6%) | 0.82 (0.37–1.81) | |
| No | 10 (23.8%) | 32 (76.2%) | Reference | |

[a]Chi-Square test.

## The prevalence of *S. pneumoniae*

*Streptococcus pneumoniae* colonization was found in 49 of 233 (21%) two-month-old infants. The youngest subject who carried *S. pneumoniae* in the nasopharynx was 46 days old and in the control group. This subject carried serotype 3 in the first nasopharyngeal swab, then 6A/6B in the second nasopharyngeal swab, 6B in the third nasopharyngeal swab, and was negative at 18 months of age.

The prevalence of *S. pneumoniae* carriage increased with age (Fig 2), with no significant difference between the control group and vaccine group ($p > 0.05$). The prevalence in 12-month-old subjects positive for *S. pneumoniae* was significantly higher than in two- month-olds in the control and vaccine groups ($p < 0.01$).

## VT and NVT serotypes prevalence by age

Fig 3 shows the prevalence of VT and NVT serotypes in both control and vaccine groups at 2, 4,12, and 18 months old. There was no significant difference between both groups at 2, 4, and 12 months old ($p > 0.05$). At 18 months old, the prevalence of VT serotypes in the vaccine group was significantly lower than the control group ($p < 0.01$). Furthermore, we observed participants in the control group who were non-pneumococcal carriers at age 2 months have a higher prevalence of VT carriage at age 12 months, compared with participants in the vaccine group who were non-pneumococcal carriers at age 2 months.

## The distribution of *S. pneumoniae* VT and NVT serotype

The most common *S. pneumoniae* VT serotypes for both groups were 6A/6B, 19F, 23F, and 14 in all age groups. In the control group at two months of age, the most common serotype was

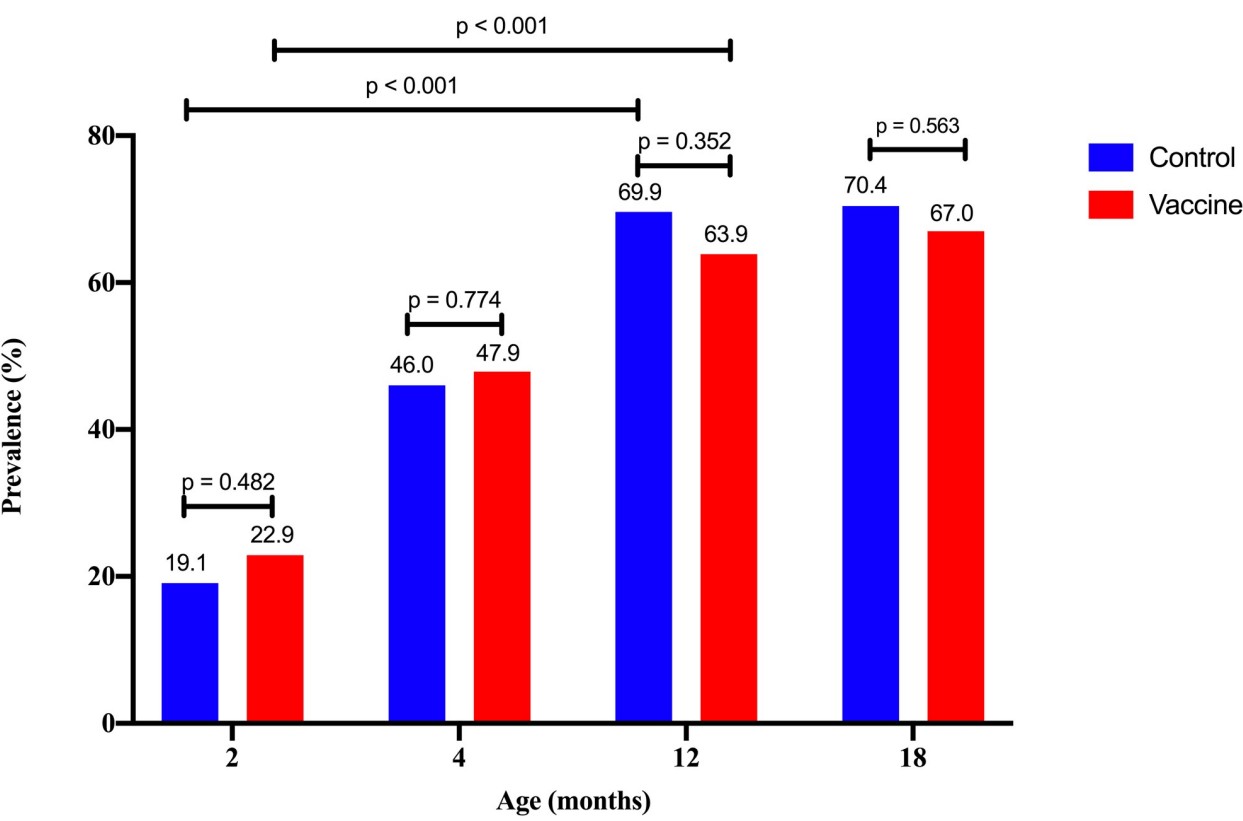

**Fig 2. The prevalence of *S. pneumoniae*-positive subjects by age in control and vaccine groups.**

14 (4/9) followed by serotype 3 (2/9), whereas in the vaccine group, it was serotype 19F (3/7) followed by 6A/6B (3/7). The prevalence changed over time. Serotypes 19A, 1, and 5 were not found in the control group at two months of age. Serotype 19A appeared later at four months of age, while serotype 1 and 5 were present at 18 months of age. In the vaccine group, serotypes 3, 19A, and 5 were not found at two and four months of age. Serotypes 3 and 19A were present at 12 months of age, while serotype 5 appeared at 18 months of age in a small number of subjects. The most common NVTs in the control group were serotypes 15B/15C, 16F, 34, and 13, whereas, in the vaccine group, the most common were 15B/15C followed by 6C/6D, 23A, and 13.

## Discussion

The study aimed to assess the effectiveness of PCV13 administration in altering nasopharyngeal *S. pneumoniae* colonization using two primary doses (in a one-month interval) and one booster dose in healthy infants and toddlers aged 2–18 months in Indonesia. We found that the sociodemographic data in both groups were relatively equal as baseline, except in nutritional status. The vaccine groups were significantly more likely than control groups to be malnourished (underweight or overweight/obese). PCVs have been found to have lower effectiveness among malnourished children. The explanation for this effect remains unknown [19, 20].

We found that the rates of *S. pneumoniae* colonization in this study was slightly higher than in other studies conducted in Indonesia. A study from Bandung in 2005 showed that the

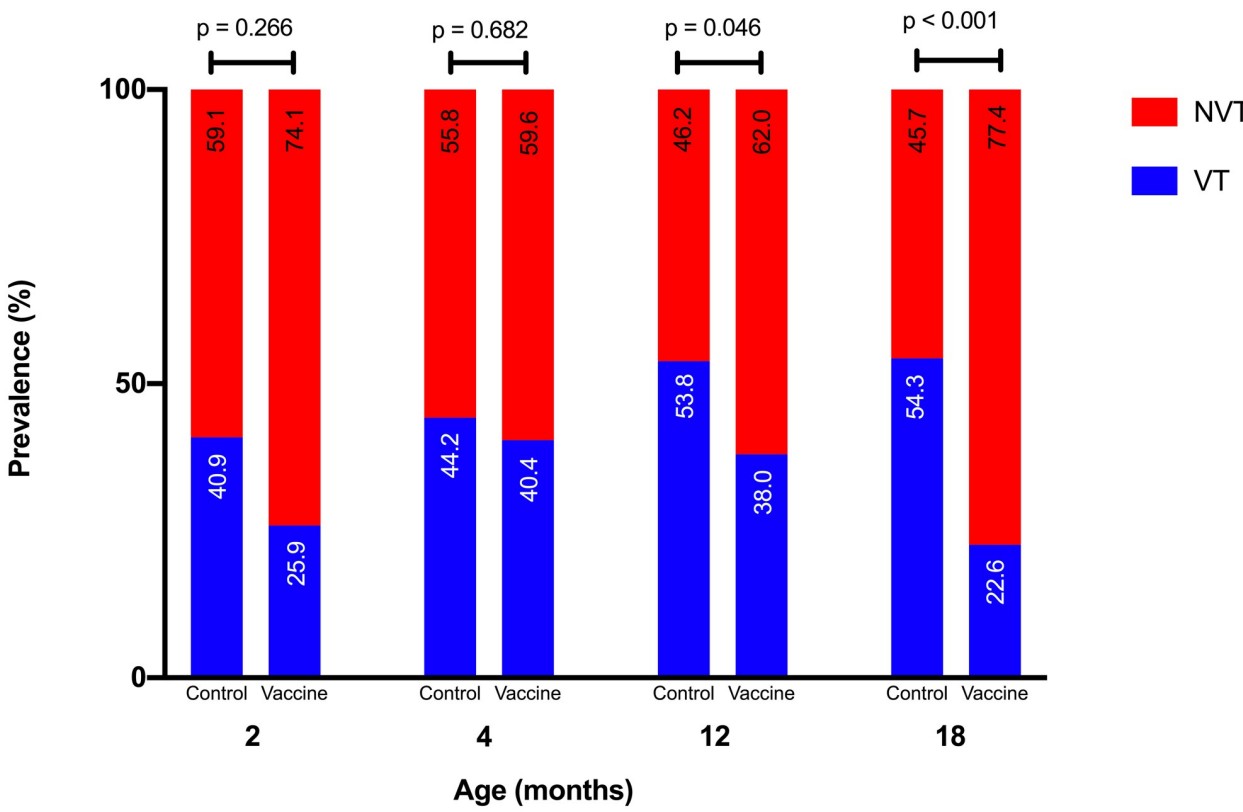

**Fig 3. VT and NVT serotype prevalence by age in control and vaccine groups.**

prevalence of pneumococci at six weeks was 12% and 13.9% at eight weeks old [21]. Murad et al. [10], in the same city, found that the prevalence increased to 15% within nine years, but was still lower than in this study. This difference may be influenced by people's behavior patterns, geographical factors, exposure to cigarette smoke, and other factors. In the Murad study, 63% of subjects were exposed to cigarette smoke, whereas in this study, 78–86.1% had been exposed. The prevalence of subjects who experienced pneumococcal colonization in this study was also higher than those found in studies in Finland (9%) and Turkey (9.7%) [22, 23]. However, the prevalence was still lower than in subjects six weeks of age in Ethiopia in 2013 (32.7%) [24].

The prevalence of *S. pneumoniae* increased with age in the control and vaccine groups, and there was no significant difference in either group in changes of *S. pneumoniae* colonization by age. Although before PCV13, the vaccine group had a higher prevalence of pneumococcal colonization than the control group, after nine months (from the second dose), it was slightly lower than in the control group and persisted up to six months after the booster administration, although without a statistically significant difference. Furthermore, there are other factors beyond PCV vaccination which affect pneumococcal colonization in the nasopharynx of infants and children, such as immune system maturity, physical activity, number of household members, frequency of respiratory tract infections, exposure to cigarette smoke, and others [25].

These findings are supported by other published studies. Heinsbroek et al. [26] stated that three years after the introduction of PCV13 in 2011 in Malawi, pneumococcal prevalence rates

remained high at 68%, whereas initially at six weeks of age, the prevalence was 43.8%. Meanwhile, an Ethiopian study by Sime et al. [24] demonstrated the prevalence of pneumococcal colonization remained high (49.1%) at two years of age after the administration of PCV10 using a 3+0 schedule when the initial pneumococcal prevalence rate at six weeks of age was 32.7%.

The persistence of nasopharyngeal pneumococcal rates a few years after the introduction of the PCV vaccine did not follow the expected trend of a higher prevalence of VT than NVT serotypes. Pneumococcal colonization was dominated by NVT serotypes with a decreasing prevalence of VT serotypes after the completion of the PCV immunization. In Ethiopia, Sime et al. [24] found that the rate of VT was 18.4% at six weeks of age and decreased to 7.1% at two years of age after PCV10 immunization. Likewise, Heinsbroek et al. [26] reported three years after the introduction of PCV13 in 2011 in Malawi, the rate of VT colonization decreased from 13% to 9.1% at 18 weeks, slightly increased to 16.5% at 1–4 years of age and stably decreased (7.9% at five years of age) after PCV13 immunization. A microsimulation model from Finland predicted elimination of VT carriage among vaccinated individuals within 5–10 years of PCV introduction [27].

We found four VT serotypes that always colonized the nasopharynx since before PCV13 immunization until after completion of the PCV13 schedule: 6A/6B, 19F, 23F, and 14. Serotypes 6A/6B, 23F, and 19F were found predominantly at the same location in Central Lombok in the previous study by Hadinegoro et al. [9] in 2011. This means that after seven years and even after PCV13 immunization on a 2+1 schedule, the dominance of these three serotypes in Lombok did not change. The Pneumococcal Conjugate Vaccine Review of Impact Evidence (PRIME) [28] stated there is a persistent carriage of VT even after more than three years of post-PCV13 administration.

Serotype 6A/6B was consistently dominant in the control and vaccine groups, both before and after completion of the PCV13 injection. It appears that PCV13 has begun to play a role in suppressing serotype 6A/6B colonization after the second dose, while for those who were unimmunized with PCV13, the prevalence of serotypes 6A/6B will continue to increase by age. The consistent presence of serotypes 6A/6B in the nasopharynx is understandable because both serotypes 6A and 6B are 2 of the 3 serotypes with the longest acquisition duration among all pneumococcal serotypes. Serotype 6A has a colonization duration of 237 days and 6B, 184 days [29].

The persistence of serotypes 6A/6B, 23F, 19F, and 14 six months after the third dose of the PCV13 administration (at 18 months old) is supported by the results of Sime et al. [24] Song et al. [30] reported that serotype 14 is highly invasive, whereas 6A, 6B, and 23F were generally less invasive, as stated in most studies. There are different clinical impacts of each serotype, such as serotype 19F, which is associated with higher mortality. Serotypes 6A/6B became the most common etiology of pneumococcal meningitis (40%) in Ugandan children under five years of age, according to their surveillance study, followed by serotypes 22A, 23F, 14, and 19A [31]. A meta-analysis study showed that serotype 14 was the most prevalent etiology of pneumococcal community-acquired pneumonia [32].

This study did not show a reduction in VT serotypes either one month or eight months after the administration of the second primary dose. The decrease in VT serotypes occurred after booster administration. A short period of the PRIME observation [28] showed that 2+1 provided a higher antibody response for most serotypes compared to the 3+0 schedule, but with no statistically significant differences, except for serotype 6B. The administration of the booster dose provides a higher antibody concentration compared to only primary doses. The PRIME [28] also showed that the 2+1 schedule provides a greater reduction of VT serotypes compared to 3+0, although this reduction was not statistically significant. However, Whitney

et al. [11] mentioned that the effectiveness between schedules only relevant in the early PCV program and context dependent. Furthermore, Flasche et al. [33] and Choi et al. [34] stated that it may even be sufficient to move to a 1+1 schedule in mature PCV program.

Goldblatt et al. [35] stated that the two-month interval provided a better immunological response than the one-month interval if immunization was started at two months of age. Whitney et al. [11] showed in a review that the IgG geometric mean concentrations of two primary doses using the two-month interval was higher than with the one-month interval, but the difference was not statistically significant. The PRIME [28] could not assess the impact of the interval schedule on reduced nasopharyngeal carriage due to a lack of data. Our study showed that two primary doses administered with a one-month interval could reduce the VT serotype compared to the control group in nasopharyngeal carriage.

The Indonesian Ministry of Health decided to implement the 2+1 schedule for PCV immunization, given simultaneously with the pentavalent vaccine (DTP/hepatitis-B-Hib & OPV) in the National Immunization Program, according to Indonesia Technical Advisory Group on Immunization recommendation. This schedule will increase the immunization coverage and compliance of the parent due to a reduction in visits to the health facility.

## Study limitations

The limitation of this study is the difficulty in determining the duration of serotype acquisition of nasopharyngeal *S. pneumoniae* due to a wide range and inconsistent interval of nasopharyngeal swab collection. In addition, this study includes relatively small sample size and only take place in single geographic area with distinct pneumococcal epidemiology based on national demonstration program, which may limit application to broader areas/populations in Indonesia but will help to evaluate the program. The logistic regression models were also tried to be performed to investigate the potential confounders, but the analysis as described did not account for potential confounders.

## Conclusions

This study shows lower prevalence of VT and higher prevalence of NVT serotypes among children who were PCV13 2+1 vaccinated compared with those who were unvaccinated. Another notable finding is that serotypes 6A / 6B in the nasopharynx, as the most dominant serotypes found in infants, were lower among children who were PCV13 vaccinated. The result from this study will be the beginning of future vaccine evaluation. Further studies are needed in larger population and longer period of study.

## Supporting information

**S1 Table. Vaccine type serotype distribution by age.**
(DOCX)

**S2 Table. Non-vaccine type serotype distribution by age.**
(DOCX)

## Acknowledgments

We would like to express our gratitude to all staff from the Ministry of Health in West Nusa Tenggara, Biomedical Laboratory General Hospital of West Nusa Tenggara Province, Eijkman Institute for Molecular Biology for the collection and processing of study data, and all research assistants who have supported throughout the study processes.

## Author Contributions

**Conceptualization:** Sri Rezeki S. Hadinegoro.

**Data curation:** Dodi Safari, Miftahuddin Majid Khoeri.

**Formal analysis:** Julitasari Sundoro.

**Investigation:** Dodi Safari, Miftahuddin Majid Khoeri.

**Methodology:** Joedo Prihartono.

**Resources:** Anis Karuniawati, Julitasari Sundoro.

**Supervision:** Bambang Supriyatno, Zakiudin Munasir, Anis Karuniawati, Sri Rezeki S. Hadinegoro, Dodi Safari.

**Writing – original draft:** Ari Prayitno.

**Writing – review & editing:** Zakiudin Munasir, Sri Rezeki S. Hadinegoro.

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
