## [Decision Letter · Decision Letter 0]

27 Aug 2020

PONE-D-20-14898

Dynamic changes in nasopharyngeal pneumococcal serotype before and after pneumococcal comjugate vaccine administered on a 2+1 schedule

PLOS ONE

Dear Dr. Prayitno,

Thank you for submitting your manuscript to PLOS ONE. After careful consideration, we feel that it has merit but does not fully meet PLOS ONE’s publication criteria as it currently stands. Therefore, we invite you to submit a revised version of the manuscript that addresses the points raised below during the review process.

We look forward to receiving your revised manuscript.

Kind regards,

Ray Borrow, Ph.D., FRCPath

Academic Editor

PLOS ONE

Journal Requirements:

Reviewers' comments:

Reviewer's Responses to Questions

**Comments to the Author**

1. Is the manuscript technically sound, and do the data support the conclusions?

Reviewer #1: No

Reviewer #2: Partly

Reviewer #3: Yes

2. Has the statistical analysis been performed appropriately and rigorously? 

Reviewer #1: No

Reviewer #2: I Don't Know

Reviewer #3: Yes

3. Have the authors made all data underlying the findings in their manuscript fully available?

Reviewer #1: No

Reviewer #2: No

Reviewer #3: Yes

4. Is the manuscript presented in an intelligible fashion and written in standard English?

Reviewer #1: Yes

Reviewer #2: Yes

Reviewer #3: Yes

5. Review Comments to the Author

Reviewer #1: This paper aims to assess changes of pneumococcal carriage pre and post PCV13 administration, as a 2+1 schedule, in Indonesia. The key findings were that PCV13 as a 2+1 schedule reduced VT serotype carriage and reduced carriage of 6A/B as dominant serotypes. Overall, the scientific background and rationale for the investigation being reported could be strengthened and given further specificity. Strength of rationale would benefit is lacking. Overall, there is insufficient information regarding methods (e.g., sample size calculation, inclusion / exclusion criteria). It would be helpful to include further details in the results, such as a flow chart of participants, and a tabular comparison of participant characteristics stratified by control and vaccine cohorts. Largely the strengths and limitations of the paper and unaddressed, and should be reflected upon. In addition to the strong suggestion of using the STROBE checklist for observational studies, specific comments are below:

The study design is a prospective cohort – it would be of benefit to include this in the Title.

1. Title: Page 1: This title is descriptive and would draw the attention of those working in infectious diseases epidemiology, vaccinology, public health, and microbiology. Those working in the pneumococcal field would be drawn in particular. I would recommend specifying which PCV is under investigation. Further, it would be helpful to include the study design in the title. For example, something like “Pneumococcal nasopharyngeal carriage in Indonesian infants and toddlers post-PCV13 vaccination in a 2+1 schedule: a prospective cohort study.”

2. Abstract Page 2 Methods: Would suggest inclusion of a few more details in the Methods section, including how participants were recruited; how pneumococci were detected and isolated; how was dominance of serotypes determined; and what statistical methods were used for inferential analyses?

3. Abstract Page 2 Methods: Here and throughout the paper, would recommend using the term prevalence if carriage is reported as % in this paper, rather than proportion. A proportion can only have values from zero to one.

4. Abstract Page 2 Methods: Would recommend including the age eligibility of participants and any other broad inclusion criteria – e.g., this study recruited infants 2 months of age, followed until 18 months of age.

5. Here and throughout the paper, would recommend using the term prevalence if carriage is reported as % in this paper, rather than proportion. A proportion can only have values from zero to one.

6. Abstract Page 2 Results: Would suggest including a summary key demographics (e.g., x% of the participants were female), and any differences in participant characteristics by control and vaccine groups.

7. Abstract Page 2 Results: Would suggest including units and % after each reported age and prevalence, i.e., “the [prevalence] of VT serotypes in control groups aged 2 months, 4, months, 12 months, and 18 months was 40.9%, 44.2%, 53.8%, and 54.3%, respectively.

8. Abstract Page 2 Results: How was the p value determined, with regard to the reported significant decrease in VT serotypes compared with NVT serotypes? Further, would recommend reporting the p value itself, rather than p<0.05. In addition, it would be expected that the decline in VT serotypes would be greater than any decline in NVT serotypes, given the introduction of PCV13.

9. Abstract Page 3 Conclusion: I am unsure that the stated conclusion can be drawn from what appear to be crude analyses.

10. Introduction Page 3: Here and throughout, references are not cited where required – e.g., a reference for the statement that pneumonia is the leading cause of death in children under five years old worldwide – could recommend McAllister et al The Lancet, 2018 available here: https://www.thelancet.com/journals/langlo/article/PIIS2214-109X(18)30408-X/fulltext.

11. Introduction Page 3: Would recommend editing “The most common cause of pneumonia is Streptococcus pneumoniae” to “A leading cause of pneumonia is Streptococcus pneumoniae”.

12. Introduction Page 3: Would suggest rewording “PCV administration can change …may be relative unchanged”. PCV reduces carriage, and thereby transmission of VT pneumococci, providing both direct and indirect protection from common disease causing serotypes. As a PCV program matures, serotype replacement may occur.

13. Introduction Page 3 and 4: Would suggest editing “In Bandung City, Indonesia, the pneumococcal carriage prevalence was 22.0% at the beginning of the study and increased to 68.4% after 10 months” to reflect that reported prevalence includes data from children recruited from both Bandung city and from a semi-rural area in Padalarang district ~25km from Bandung city – this will help provide some further context for the reader.

14. Introduction Page 4: In the paragraph describing the recommendations of the WHO with regard to schedules, it would be useful to include the information provided on page 13 as to where and why the WHO recommends different schedules. This will help provide background and information as to the current study setting.

15. Introduction Page 4: To provide some further specificity, would recommend indicating that “the goal of this study is to assess the effectiveness of PCV13” rather than just PCV; and including the age of healthy children; similarly, would recommend indicating the age of children, e.g., “… in healthy children aged 2 – 18 months in Indonesia.”

16. Introduction: were there any prespecified hypotheses?

17. Introduction Page 4: To provide some further specificity, would recommend indicating that “the goal of this study is to assess the effectiveness of PCV13” rather than just PCV; and including the age of healthy children; similarly, would recommend indicating the age of children, e.g., “… in healthy children aged 2 – 18 months in Indonesia.”

18. Methods Page 4, Study Design and Participants: The description of the study design, indication of location, and time frame of study were very clear. Further, the overt statement of ethical approval for the study is important, and described clearly.

19. Methods Page 4/5, Study Design and Participants: Some further details describing the setting and location would be welcome – for example, the population size, whether access to healthcare is universal, and rates of pneumonia among children aged under 5

20. Methods Page 4/5, Study Design and Participants: The description of the study design, indication of setting, and time frame of study were very clear. Further, the overt statement of ethical approval for the study is important, and described clearly.

21. Methods, Study Design and Participants Page 5: Some key information regarding participants would be of value to include. For example, the eligibility criteria, sources and methods of participant recruitment, and methods of follow-up. Were control and vaccine groups matched? It appears that they may be age matched – where there any other matching criteria if relevant?

22. Methods, Study Design and Participants Page 5: The paper states that a consecutive sampling method was used, i.e., one in which every subject meeting inclusion criteria is selected until the required sample size is achieved. However, no sample size justification / calculation is provided, and neither are inclusion or exclusion criteria. Therefore it is unknown if the sample size provides sufficient power for this study. Further, if “healthy” children were recruited, what were the exclusion criteria – i.e., how was “healthy” determined? Were temperatures assessed and any child with a fever excluded? Were children with comorbidities, AOM, respiratory tract infections, taking antibiotics excluded? If a recruited child became unwell over the period they aged from 2 months to 18 months were they excluded?

23. Methods – would recommend a section that describes and defines all variables. That is, define clearly, all outcomes, exposures, predictors, potential confounders, and effect modifiers as relevant. For example, “VT carriage was defined as carriage of serotypes 1, 3, 4, 5, 6A, 6B, 7F, 9V, 14, 19A, 19F, 18C, and 23F”.

24. Methods, Sample collection, page 5: would recommend specifying the swab material – presumably flocked nylon, as opposed to just flocked.

25. Methods, Sample collection, page 5: would recommend reviewing and referencing Satzke et al, Vaccine 2013 available here: https://www.clinicalkey.com.au/#!/content/playContent/1-s2.0-S0264410X13011742?returnurl=https:%2F%2Flinkinghub.elsevier.com%2Fretrieve%2Fpii%2FS0264410X13011742%3Fshowall%3Dtrue&referrer=https:%2F%2Fpubmed.ncbi.nlm.nih.gov%2F24331112%2F with regard to microbiological methods. It is important to cite this reference regarding the methods of NP sampling being in accordance with WHO guidelines.

26. Methods, Laboratory procedures, page 6: How was the dominant serotype determined?

27. Methods, Laboratory procedures, page 6: What kind of blood was used in the blood agar plates?

28. Methods, Data analysis page 6: Was it possible to type all isolates? Were there any instances where an isolate was detected, but typing was not possible due to biological reasons or technical issues? If so, how was this missing data handled?

29. Methods, Data analysis page 6: It is unclear why participants with multiple carriage were assigned either VT or NVT only. Where participants carry both VT and NVT, it should be possible to record an observation of carriage for both VT and NVT serotypes.

30. Methods, Data analysis page 6: It is unclear how the statistical methods employed address the stated goal of assessing vaccine effectiveness of PCV13 as a 2+1 schedule in health Indonesian children. It seems crude analysis only has been undertaken. Is this because the sample size was insufficient to permit more robust analysis, such as the standard 1 – odds ratio multiplied by 100, to determine VE? Further, would recommend that the analysis takes into account the significant differences in terms of participant characteristics by vaccine and control group, to determine the odds ratio for such analysis.

31. Methods, Data analysis page 6: It is important to describe any efforts made to reduce any potential sources of bias – these may arise through selection processes, measurement processes, and differences in groups. Please could it be stated whether the microbiologists undertaking laboratory procedures were blinded to participant cohort and demographics?

32. Methods, Data analysis page 6: Following the previous point, it is strongly recommended, that the two groups, i.e., vaccine and control, have participant demographics summarized via standard methods (i.e., categorical variables by counts and percentages, continuous variables by mean and standard deviation or median and interquartile range, as appropriate to the distribution) and then compared using appropriate methods. This is because it is important to determine if control and vaccine groups were different at baseline – differences in outcome may be due to differences at baseline, rather than the exposure of interest, and this should be considered.

33. Methods: It is important to describe any efforts made to reduce any potential sources of bias – these may arise through selection processes, measurement processes, and differences in groups.

34. Methods: Which software was used for databases, and for descriptive and inferential analyses?

35. Results: demographic characteristics, page 7: Please include a flow chart of participants

36. Results: demographic characteristics, page 7: A summary of the participant characteristics by control and vaccine groups is essential. Would strongly recommend including this in tabular form, along with relevant comparisons. It would be of benefit to report characteristics such as sex distribution across groups, and any differences in characteristics by group, along with a statistical comparison.

Results: demographic characteristics, page 7: The counts and percentages reported with regard to nutritional status and exposure to cigarette smoke by vaccine and control group appear to be quite different. Both of these factors have been shown in previous studies to be associated with pneumococcal carriage. For this reason, it is important to compare participant demographics by cohort group, and then adjust as necessary in inferential analyses.

37. Results: demographic characteristics, page 7: Recommend replacing table 1 with a comparison of characteristics by cohort group, then undertaking formal analyses of association of factors associated with carriage, and using the results to determine effectives of PCV13.

38. Results: The proportion of S. pneumoniae, page 8: Here, and throughout, as in the Abstract, suggest replacing “proportion” with prevalence, and including a 95% confidence interval.

39. Results: The pattern of S. pneumoniae serotype colonization, page 9: Suggest moving “Multiple colonization is colonization by more than one ….during the entire study.” to Methods, in a section on Variables.

40. Results: The pattern of S. pneumoniae serotype colonization, page 10: This paper states that “After vaccination with PCV13, we expected that VT serotypes would be replaced by NVT serotypes or become negative, and did not expect that NVT serotypes would be replaced by VT or there would be no colonization by VT. “ Any such statements could usefully be reworded as pre-specified, hypotheses and included in the Introduction. However, the duration of this study is very short (16 months) which is may not be sufficient to for serotype replacement in a vaccine naïve population to emerge.

41. Results: The pattern of S. pneumoniae serotype colonization, page 10: This paper refers to “subjects with negative colonization chang[ing] to VT’ etc. It would be of value to describe carriage as observations. For example, “We observed participants in the control group who were non-pneumococcal carriers at age 2 months to have a higher prevalence of VT carriage at age 12 months, compared with participants in the vaccine group who were non-pneumococcal carriers at age 2 months.”

42. Discussion, page 10: Recommend beginning the discussion with a summary of the key results along with reference to the study objectives. For example, “this study aimed to assess the effectiveness off of PCV13 …. We found ….”

43. Discussion, page 11: This paper states that “This means that PCV13 administration on a 2 + 1 schedule did not prevent an increase in pneumococcal colonization.” It is not clear whether this study has a sufficient sample size, or statistical methods that were undertaken, that support this statement.

44. Discussion, page 11: This paper states that “These findings prove that there are other factors beyond PCV vaccination which affect pneumococcal colonization in the nasopharynx of infants and children…” This study does not examine factors associated with pneumococcal carriage. As such, it does not provide evidence to support this statement. However, many previous studies have investigated factors associated with pneumococcal carriage, including Fadlyana et al Pneumonia, 2018 (available here https://link.springer.com/article/10.1186/s41479-018-0058-1) which describes factors associated with pneumococcal nasopharyngeal carriage in young children living in Indonesia. Further, given it has been recognised that factors other than vaccination affect pneumococcal carriage, it would be worthwhile conducting analyse that a) determine factors relevant to carriage in the current study sample and b) take such factors into account in analyses.

45. Discussion, page 12: The paper reports that “PCV13 is expected the result in no colonization of VT S.pneumoniae serotypes or replacement by NVT serotype. Although a microsimulation model from Finland predicted elimination of VT carriage among vaccinated individuals within 5 – 10 years of PCV introduction (under the assumption of a 90% vaccine coverage with 50% vaccine efficacy against acquisition) (see Nurhonen et al PLOS ONE 2013 https://journals.plos.org/plosone/article?id=10.1371/journal.pone.0056079), other evidence suggests that vaccine impact from transmission models based on data from low carriage prevalence settings, may translate to high carriage prevalence settings. Further, any elimination of VT carriage is likely to occur in settings with a mature PCV program, rather than 16 months of PCV, as supported by the paper’s inclusion of reference to the PRIME finding that there is persistent carriage of VT even after more than three years of post-PCV13 administration. As such, it is inconsistent to expect “PCV13 vaccination …to result in no colonization of VT S.pneumoniae serotypes…”

46. Discussion, page 13: I am unconvinced that this study provides evidence to support the statement that “two primary doses are not enough to reduce VT serotypes in nasopharyngeal colonization.”

47. Discussion, page 13: The paper states that a “booster dose will only be effective if it is after two primary doses.” It should be noted that this is likely context dependent. For example, in a setting with a mature PCV program, it may be possible to move to a 1+ 1 schedule, as the UK has done. See Flasche et al PLOS Mid, 2015 (https://journals.plos.org/plosmedicine/article?id=10.1371/journal.pmed.1001839) and Choi et al PLOSMed, 2019 (https://journals.plos.org/plosmedicine/article?id=10.1371/journal.pmed.1002845).

48. Study limitations, page 14: I agree that a limitation of this study is the difficulty in determining duration of serotype acquisition – more frequent swabbing would be necessary to achieve this. Other limitations should also be acknowledged and reflected upon, including: the lack of generalizability beyond other healthy children; the differences between control and vaccine groups; reflections upon the sampling method; if the sample size is of sufficient size to provide power.

49. Conclusions, page 14: The effectiveness of 2+1 studies is known. This study has not provided evidence for this, and does not assesses vaccines.

50. It may be of use to read and reflect upon the following articles:

a. Wahl et al, Lancet Glob Health 2018 https://www.thelancet.com/journals/langlo/article/PIIS2214-109X(18)30247-X/fulltext

b. Fadlyana et al Pneumonia, 2018 https://link.springer.com/article/10.1186/s41479-018-0058-1

c. Satzke et al, PLOS Med, 2015 https://journals.plos.org/plosmedicine/article?id=10.1371/journal.pmed.1001903

d. Greenberg et al, Clin Infect Dis 2016 https://academic.oup.com/cid/article/42/7/897/322584

e. Farida et al, PLOS ONE 2014 https://journals.plos.org/plosone/article?id=10.1371/jousrnal.pone.0087431

Reviewer #2: Important note: This review pertains only to ‘statistical aspects’ of the study and so ‘clinical aspects’ [like medical importance, relevance of the study, ‘clinical significance and implication(s)’ of the whole study, etc.] are to be evaluated [should be assessed] separately/independently. Further please note that any ‘statistical review’ is generally done under the assumption that (such) study specific methodological [as well as execution] issues are perfectly taken care of by the investigator(s). This review is not an exception to that and so does not cover clinical aspects {however, seldom comments are made only if those issues are intimately / scientifically related & intermingle with ‘statistical aspects’ of the study}. Agreed that ‘statistical methods’ are used as just tools here, however, they are vital part of methodology [and so should be given due importance].

COMMENTS: Study design is fairly simple; however, I have certain questions/doubts: In ‘Methods’ section, ‘Study design and participants’ subsection you said “Control group members were not immunized for PCV and lived in Central Lombok, while vaccine group members lived in West or East Lombok and received PCV13 immunization” which is O.K., but later you say “We use consecutive sampling method”, how is that done? What exactly you want to convey?

There are no comments on ‘Sample collection’ and ‘Laboratory procedures’ because they are beyond my area of knowledge. ‘Data analysis’ again is very simple [like simple study design]. You say ‘the Kolmogorov-Smirnov test was performed when some assumptions of chi-square were not satisfied’. The question is ‘which assumptions (of chi-square) were not satisfied?’ [note that Kolmogorov-Smirnov test is only a goodness-of-fit test; whereas chi-squared is a goodness-of-fit test as well as a test of association]

If only ‘0’ frequency in two cells [table-1] is the issue, I recommend to use RIDIT analysis [Bross IDJ. ‘How to use RIDIT analysis’. Biometrics, 1958; 14:18-38. There are many recent references, but they are on application results. This is (an original, old classic and) gives details on ‘how to use’ the technique].

Since the ‘Article Type’ is ‘Clinical Trial’ it is expected to give details of required sample size estimation. Even ‘allocation’ procedure is desirable to be describe adequately. As said in ‘Methods’ section of abstract it is actually the ‘prospective cohort study’, but it is still desirable to discuss about ‘required sample size estimation’. I am not sure regarding important ‘clinical significance’ and ‘implication(s)’ of this study and so in my opinion it is vital to assess the study in this context by subject expert. ‘What the study contributes?’ is always the important question.

‘Study limitations’ mentioned are not very clear. As said earlier ‘Study design’ as well as ‘Data analyses’ are simple and therefore, not much to comment.

Reviewer #3: The authors present the results of a study investigating nasopharyngeal colonization rates with vaccine and non-vaccine type pneumococci following vaccination of infants with PCV in Indonesia, with the finding of lower VT colonization rates in the vaccinated group at 18 months. As mentioned by the authors, pneumonia and pneumococcal infections are a major cause of global mortality among infants and young children, and such studies are important to understand pneumococcal epidemiology and inform immunization programs in different regions.

Major comments:

1. Revise/reorganize abstract and body of manuscript to clarify the relationship between the present study (which is referred to as a prospective cohort) and the clinical trial with vaccine and control groups. Was NP colonization a prespecified endpoint in a clinical trial, or was this a separate/ancillary cohort study of clinical trial participants?

2. A table or figure depicting pneumococcal serotype distribution would be helpful, either in the main manuscript or as supplemental material

3. Limitations also appear to include relatively small sample size and single geographic area with distinct pneumococcal epidemiology, which may limit application to broader areas/populations

Minor comments:

1. Title-spelling of pneumococcal conjugate vaccine

2. Introduction-clarify if pneumococcus is the most common cause of bacterial pneumonia (viruses generally more common overall)

6. PLOS authors have the option to publish the peer review history of their article (what does this mean?). If published, this will include your full peer review and any attached files.

Reviewer #1: No

Reviewer #2: No

Reviewer #3: No

---

## [Author Response · Author response to Decision Letter 0]

15 Oct 2020

I have given some respond to reviewers as I attached in respond to reviewer file that I have submitted

---

## [Decision Letter · Decision Letter 1]

4 Nov 2020

PONE-D-20-14898R1

Pneumococcal nasopharyngeal carriage in Indonesia infants and toddlers post-PCV13 vaccination in a 2+1 schedule: a prospective cohort study

PLOS ONE

Dear Dr. Prayitno,

Thank you for submitting your manuscript to PLOS ONE. After careful consideration, we feel that it has merit but does not fully meet PLOS ONE’s publication criteria as it currently stands. Therefore, we invite you to submit a revised version of the manuscript that addresses all of the points raised below during the review process.

We look forward to receiving your revised manuscript.

Kind regards,

Ray Borrow, Ph.D., FRCPath

Academic Editor

PLOS ONE

Reviewers' comments:

Reviewer's Responses to Questions

**Comments to the Author**

1. If the authors have adequately addressed your comments raised in a previous round of review and you feel that this manuscript is now acceptable for publication, you may indicate that here to bypass the “Comments to the Author” section, enter your conflict of interest statement in the “Confidential to Editor” section, and submit your "Accept" recommendation.

Reviewer #1: (No Response)

Reviewer #2: All comments have been addressed

Reviewer #3: (No Response)

2. Is the manuscript technically sound, and do the data support the conclusions?

Reviewer #1: No

Reviewer #2: (No Response)

Reviewer #3: Partly

3. Has the statistical analysis been performed appropriately and rigorously? 

Reviewer #1: No

Reviewer #2: (No Response)

Reviewer #3: No

4. Have the authors made all data underlying the findings in their manuscript fully available?

Reviewer #1: Yes

Reviewer #2: (No Response)

Reviewer #3: Yes

5. Is the manuscript presented in an intelligible fashion and written in standard English?

Reviewer #1: No

Reviewer #2: (No Response)

Reviewer #3: Yes

6. Review Comments to the Author

Reviewer #1: This paper aims to assess changes of pneumococcal carriage pre and post PCV13 administration, as a 2+1 schedule, in Indonesia. The key findings were that PCV13 as a 2+1 schedule reduced VT serotype carriage and reduced carriage of 6A/B as dominant serotypes. The scientific background and rationale for the investigation being reported has been strengthened. The flow chart of participants is very useful, as is the description of the setting. The tabular comparison of participant characteristics stratified by control and vaccine cohorts is a welcome inclusion. Further specificity regarding methods would be very helpful – for example statistical methods and sample size calculation. Of concern is the appropriateness of the statistical methods to address the stated aim of the study. Some specific comments and concerns are outlined below.

1. Abstract Page 3 Conclusion: I remain unsure that the stated conclusion can be drawn from these analyses. It may be more appropriate to edit to something akin to “This study describes lower prevalence of VT and 6A/6B serotypes among children who were PCV13 vaccinated compared with those who were unvaccinated”

2. Introduction: With regard to the statement “The transmission of various pneumococcal infections can be prevented by the administration of the pneumococcal conjugate vaccine in infants under one year of age, I would recommend editing in line with the WHO definition of PCV vaccination, i.e., Two doses of PCV10 given before the age of 12 months, or one or more doses of PCV10 given at or after 12 months of age (see World Health Organization. Pneumococcal vaccines WHO position paper - 2012 - recommendations. Vaccine. 2012;30(32):4717-8. Epub 2012/05/25. doi: 10.1016/j.vaccine.2012.04.093. PubMed PMID: 22621828.]

3. Introduction: Could the authors please amend the section regarding previous research on NP carriage in children in Indonesia, to introduce the Bandung city study. This will help aid flow for the reader.

1. Introduction: The inclusion of pre-specified hypotheses in the introduction are very useful. I was unclear what the author’s meant by NVT serotypes would become “negative”. Please could this be revised?

2. Methods, Study Design and Participants Page 5: The authors have included some further details regarding sampling method, however no sample size justification / calculation has been provided. Please could the authors address this?

3. Methods, Study Design and Participants Please could the authors provide some further clarity regarding eligibility, inclusion, and exclusion criteria?

4. Methods, Data analysis page 6: Could the authors please clarify how participants with multiple carriage were recorded? For example, did the authors record observations of positive to VT and NVT serotypes if participants carried both? Further, did the authors record observations of positive to serotype specific carriage where multiple serotypes were carried by the same participant?

5. Methods, Data analysis page 6: It is unclear how the statistical methods employed address the stated goal of assessing vaccine effectiveness of PCV13 as a 2+1 schedule in health Indonesian children. It seems crude analysis only has been undertaken. Is this because the sample size was insufficient to permit more robust analysis, such as the standard 1 – odds ratio multiplied by 100, to determine VE? Further, would recommend that the analysis takes into account the significant differences in terms of participant characteristics by vaccine and control group, to determine the odds ratio for such analysis. It is not necessary to measure the antibody level or decreasing of disease burden to conduct such analysis. Some further clarity regarding how demographic characteristics were summarized would be of value. For example, “categorical variables were summarized by counts and percent, and continuous variables were summarized by median and (IQR).”

6. Methods. The authors describe a prospective cohort study. Factors affecting recruitment and enrolment of participants into a prospective cohort study are unlikely to introduce selection bias. This is because in order for selection bias to occur, the selection has to be related to both exposure and outcomes. But in a prospective cohort study, participants are enrolled before the outcome of interest has occurred (in this case, carriage). Although enrolment may be related to exposure status, the prospective nature of the study makes it difficult to consider the outcome (carriage) as influencing enrolment.

7. Results: Useful to see a comparison of characteristics by cohort group. It would be useful to include analysis of association of factors associated with carriage, and using the results to determine effectives of PCV13 – this could be achieved by building logistic regression models to investigate the association of PCV13 vaccination with the outcome of interest (carriage) adjusting for potential confounders, and then estimating vaccine effectiveness using 1 – odds ratio x 100

8. Discussion; This paper states that “This means that PCV13 administration on a 2 + 1 schedule did not prevent an increase in pneumococcal colonization.” It is not clear whether this study has a sufficient sample size, or statistical methods that were undertaken, that support this statement.

9. Discussion: I remain unconvinced that this study provides evidence to support the statement that “two primary doses are not enough to reduce VT serotypes in nasopharyngeal colonization”.

10. Discussion, page 13: The paper states that a “the booster dose will only be effective if it is after two primary doses and will not be effective after three primary doses.” The authors indicate that this statement is not from their current study, but from the PRIME study. In the paper cited (Whitney et al Pediatr Infect Dis J 2014), the authors state “overall booster doses are clearly beneficial for programs that use only 2 primary doses, but the clinical benefit of a booster dose remains uncertain for programs that achieve high coverage with 3 primary doses”. This is different from suggesting that a booster dose will only be effective after two primary doses and will not be effective after three primary doses. Effectiveness of booster doses and primary doses are context dependent. In a setting with a mature PCV program, it may be possible to move to a 1+ 1 schedule, as the UK has done. See Flasche et al PLOS Mid, 2015 (https://journals.plos.org/plosmedicine/article?id=10.1371/journal.pmed.1001839) and Choi et al PLOSMed, 2019 (https://journals.plos.org/plosmedicine/article?id=10.1371/journal.pmed.1002845). I would recommend some amendment to reflect the context specific nature of the effectiveness of the number of primary and booster doses.

Reviewer #2: COMMENTS: Since most of the comments (atleast major ones) made on earlier draft by me (and hopefully by other respected reviewers also as there were many) are attended positively/adequately, and, in my opinion, the manuscript is improved a lot. I feel that now the manuscript has achieved acceptable level of our journal.

Reviewer #3: With some additional clarification of the study design and characteristics of the study cohorts, several additional questions arise that should be addressed:

1. The authors now refer to the sampling method as a “convenience sample” and provide some information regarding the initial clinical trial. However key details regarding the selection methods are still lacking. Specifically:

a. How many participants were included in the original clinical trial? Recommend adding the numbers enrolled in the initial vaccine trial to the top of the flow diagram shown in Figure 1.

b. Based on the first version of the manuscript and subsequent revision, it can be inferred that consecutive participants in the clinical trial were enrolled into the two cohorts based on geographic location/vaccine group assigment, however this is still not explicitly stated. Is this accurate? Please make selection methods clear and specific in the manuscript.

c. Please include more details regarding the power analysis used to determine the size of the cohorts in this study. What endpoint(s) was the study powered for? This seems important in determining whether the study was sufficiently powered to support the authors' conclusions re: overall carriage rates.

2. While additional demographic information is now included, the analysis does not take potential confounders into account, and is, as mentioned by others, a crude analysis. For example, the characteristics of the cohorts indicate a difference in rates of malnutrition between groups-while the authors state that baseline differences could affect outcome measures (and that there is a previously reported association between pneumococcal carriage and nutritional status), the statistical models do not account for such variables as covariates. Have the authors attempted to perform a multivariate analysis to investigate this?

Minor comments:

1. Page 4: Where mentioned, provide a brief definition of serotype replacement and the general timeframe in which it typically occurs, for readers who may not be familiar with this phenomenon.

7. PLOS authors have the option to publish the peer review history of their article (what does this mean?). If published, this will include your full peer review and any attached files.

Reviewer #1: No

Reviewer #2: No

Reviewer #3: No

---

## [Author Response · Author response to Decision Letter 1]

18 Dec 2020

Dear Dr./Mr./Ms. Reviewers

Thank you for giving us the opportunity to submit a revised draft of our manuscript titled Pneumococcal nasopharyngeal carriage in Indonesia infants and toddlers post-PCV13 vaccination in a 2+1 schedule: a prospective cohort study to PLOS ONE. We appreciate the time and effort that you and the reviewers have dedicated to provide us with insightful comments on our paper. We have edited the manuscript to address the suggestions provided by the reviewers. You can read our responses to all comments and questions from reviewers in the "Responses to Reviwers" file.

---

## [Decision Letter · Decision Letter 2]

8 Jan 2021

Pneumococcal nasopharyngeal carriage in Indonesia infants and toddlers post-PCV13 vaccination in a 2+1 schedule: a prospective cohort study

PONE-D-20-14898R2

Dear Dr. Prayitno,

We’re pleased to inform you that your manuscript has been judged scientifically suitable for publication and will be formally accepted for publication once it meets all outstanding technical requirements.

Kind regards,

Ray Borrow, Ph.D., FRCPath

Academic Editor

PLOS ONE

Additional Editor Comments (optional):

Reviewers' comments:

Reviewer's Responses to Questions

**Comments to the Author**

1. If the authors have adequately addressed your comments raised in a previous round of review and you feel that this manuscript is now acceptable for publication, you may indicate that here to bypass the “Comments to the Author” section, enter your conflict of interest statement in the “Confidential to Editor” section, and submit your "Accept" recommendation.

Reviewer #2: All comments have been addressed

Reviewer #3: (No Response)

2. Is the manuscript technically sound, and do the data support the conclusions?

Reviewer #2: Yes

Reviewer #3: Yes

3. Has the statistical analysis been performed appropriately and rigorously? 

Reviewer #2: Yes

Reviewer #3: Yes

4. Have the authors made all data underlying the findings in their manuscript fully available?

Reviewer #2: Yes

Reviewer #3: Yes

5. Is the manuscript presented in an intelligible fashion and written in standard English?

Reviewer #2: Yes

Reviewer #3: Yes

6. Review Comments to the Author

Reviewer #2: As said earlier, all the comments were already attended positively/adequately, now the manuscript is improved a lot. No major issue left, in my opinion.

Reviewer #3: The authors have addressed my previous comments and the manuscript is overall much improved from the original submission.

A minor item that could be clarified is the statement that multivariate analysis produced "a very big number." I'm not sure what number this refers to, for clarity it is likely sufficient to simply state as a limitation that the analysis as described did not account for potential confounders.

7. PLOS authors have the option to publish the peer review history of their article (what does this mean?). If published, this will include your full peer review and any attached files.

Reviewer #2: **Yes: **Dr. Sanjeev Sarmukaddam

Reviewer #3: No

---

## [Editor Report · Acceptance letter]

15 Jan 2021

PONE-D-20-14898R2 

Pneumococcal nasopharyngeal carriage in Indonesia infants and toddlers post-PCV13 vaccination in a 2+1 schedule: a prospective cohort study 

Dear Dr. Prayitno:

I'm pleased to inform you that your manuscript has been deemed suitable for publication in PLOS ONE. Congratulations! Your manuscript is now with our production department. 

Kind regards, 

on behalf of

Prof. Ray Borrow 

Academic Editor

PLOS ONE